# STAT1 deficiency predisposes to spontaneous otitis media

Daniel Bodmer[1], Peter Kern[1], David Bächinger[2], Arianne Monge Naldi[2], Soledad Levano Huaman[1] *

1 Department of Biomedicine and Clinic for Otolaryngology, Head and Neck Surgery, University Basel Hospital, Basel, Switzerland, 2 Department of Otorhinolaryngology, Head and Neck Surgery, University of Zurich, University Hospital Zurich, Zurich, Switzerland

* s.levano@unibas.ch

**Data Availability Statement:** All relevant data are within the manuscript.

**Funding:** The authors received no specific funding for this work.

## Abstract

Signal transducer and activator of transcription 1 (STAT1) is known to be an important player in inflammatory responses. STAT1 as a transcription factor regulates the expression of multiple proinflammatory genes. Inflammatory response is one of the common effects of ototoxicity. Our group reported that hair cells of STAT1 knockout (STAT1-KO) mice are less sensitive to ototoxic agents *in-vitro*. The effect of inflammatory responses in STAT1-KO mice has primarily been studied challenging them with several pathogens and analyzing different organs of those mice. However, the effect of STAT1 ablation in the mouse inner ear has not been reported. Therefore, we evaluated the cochlear function of wild type and STAT1-KO mice via auditory brain stem response (ABR) and performed histopathologic analysis of their temporal bones. We found ABR responses were affected in STAT1-KO mice with cases of bilateral and unilateral hearing impairment. Histopathologic examination of the middle and inner ears showed bilateral and unilateral otitis media. Otitis media was characterized by effusion of middle and inner ear that varied between the mice in volume and inflammatory cell content. In addition, the thickness of the middle ear mucosae in STAT1-KO mice were more pronounced than those in wild type mice. The degree of middle and inner ear inflammation correlated with ABR threshold elevation in STAT1-KO mice. It appears that a number of mice with inflammation underwent spontaneous resolution. The ABR thresholds were variable and showed a tendency to increase in homozygous and heterozygous STAT1-KO mice.

These findings suggest that STAT1 ablation confers an increased susceptibility to otitis media leading to hearing impairment. Thus, the study supports the new role of STAT1 as otitis media predisposition gene.

## Introduction

Otitis media is a multifactorial disease of the middle ear characterized by inflammation of the middle ear cavity and also includes Eustachian tube dysfunction. Treatment outcomes vary across the patients with similar symptoms suggesting variation in the underlying

**Competing interests:** The authors have declared that no competing interests exist.

pathophysiology probably due to genetic and environmental factors. An increasing number of candidate genes are associated to otitis media in humans and animals including *Chd7*, *Eya4*, *Enpp1*, *Fbxo11*, *Lmna*, *Mcph1*, *Oxgr1*, *Slc25a21*, *Splunc1*, *Tgif1*, *Tlr4*, *TNFA* among others [1–9]. While Tian et al. found high incidence of hearing loss in *Enpp1*$^{asj/asj}$ mice (90%), *Tlr4*-deficient mice presented around 50% cases of otitis media and *Splunc1*-deficient mice around 31% cases of otitis media [3, 6, 9]. Most of those knockout mice presented hyperplasia in the middle ear cavity with some degree of inner ear inflammation. Two independent transcriptome studies focused on pathogen induced otitis media and provided an overview of gene response during otitis media finding key genes and confirming some of those known genes associated to otitis media [10, 11].

STAT proteins are key mediators of the Janus Kinase (JAK)/STAT pathway, which is reported to be involved in subsequent transcriptional response to numerous cytokines and interferons. STAT proteins are present in the cytoplasm in inactive forms. After phosphorylation of STATs via JAK kinase, phosphorylated STATs forms dimers and translocate to the nucleus to initiate transcription of genes involved in inflammation process.

The increasing interest on STAT1 gene in the hearing research is mostly due to its participation in inflammatory processes among others. Several studies including ours have identified a number of components of the underlying mechanisms by which STAT1 regulates hair cell damage *in-vitro* and *in-vivo* [12–15]. Our early study used STAT1-KO pups to perform *in-vitro* studies and we reported that organs of Corti of STAT1-KO mice were more resistant against ototoxic drugs [14]. The immunological phenotype of STAT1-KO mice has been investigated reporting that the B lymphocytes, T lymphocytes, monocytes, macrophages and granulocytes were produced in normal numbers [16]. STAT1-KO mice showed poorly transcriptional response upon IFN stimulation, however transcriptional responses to other cytokines seems to be normal [16, 17]. To describe the infections phenotype, STAT1-KO mice were challenged with pathogens including viruses, bacteria, parasites and fungus reporting an increased susceptibility to infection (reviewed in [17]). Mice carrying macrophage-specific STAT1 ablation infected with *Cryptococcus neoformans* strain showed increased inflammation and defective macrophage activation [18]. Importantly, mutations in STAT1 gene exist in humans; patients carrying either loss or gain of function (GOF) STAT1 mutations are susceptible to infections including ear infections [19–21].

In this study, we report that STAT1-KO mice presented hearing impairment and loss of function of STAT1 resulted in inflammatory processes within the middle ear.

## Material and methods

### Mice and genotyping

STAT1-deficient mice (mixed C57BL/6-129/SvEv) lack the DNA binding domain of STAT1, and initial characterization has been described elsewhere [16]. STAT1-KO mice were kindly provided by Professor Skoda at University Hospital Basel, Switzerland. Mice were 3 to 18 weeks old. The animals were housed in pathogen-free conditions at the animal facility of the Department of Biomedicine of the University Hospital Basel. A total of 32 STAT1-KO mice and 12 wild type mice were used in this study. All animal experiments were conducted with the approval of Animal Care Committee of the Canton Basel City, Switzerland. PCR amplification of genomic DNA was used to identify mouse genotypes. The genotypes of the mice were determined after ABR measurements, except for the group of 12–13 weeks that was genotyped before ABR measurements.

## Assessment of hearing by auditory brainstem response (ABR)

ABR thresholds for click were measured in STAT1 mice aged 3 weeks (wild type, n = 1; heterozygote n = 7), 9 weeks (wild type, n = 2; heterozygotes, n = 4; homozygotes, n = 4), 12–13 weeks (wild type aged 12 weeks, n = 4; homozygotes aged 13 weeks, n = 4), and 18 weeks (wild type, n = 5; heterozygotes, n = 7, homozygotes, n = 6) as described earlier [22]. The hearing threshold were assessed using a Tucker Davis System (Tucker Davis Technologies, Alachua FL). ABR recordings were performed in anesthetized mice with an i.p. injection of a combination of ketamin (65 mg/kg), xylazine (13 mg/kg) and acepromazine (2mg/kg). During the ABR recordings, the mice were placed on a heating pad in a soundproof chamber. Stainless-steel needle electrodes were placed subcutaneously at the vertex (active), in the ipsilateral mastoid region (reference) and near the base of the tail (ground). The click stimulus generated by TDT system 3 hardware and BioSigRP system were delivered from an electrostatic speaker in a close field from 80 to 5 dB SPL in 5 dB SPL steps. The ABR wave forms were averaged over 500 repetitions and stored for further analysis. The ABR threshold was defined as the lowest stimulus at which a reliable positive waveform could be observed. Those animals that did not present any signal at any level for a click-evoked ABR suggested that their thresholds could be above 80 dB sound pressure level (SPL), which was the upper limit of stimulus presentation. For those cases with missing ABR values, we score the ABR thresholds to 85 dB SPL for subsequent analyses, even though their true thresholds may be higher.

## Ear histology

The middle and inner ears of mice at 9, 12–13 and 18 weeks of age were examined by histology. Paraffin sections of mouse ears were prepared and stained using standard histological procedures. After completion of ABR measurements mice were deeply anesthetized and the cochlea quickly removed from the skull and fixed in 4% PFA at 4˚C overnight. After decalcification in 10% EDTA, samples were embedded in paraffin (TP1020 benchtop tissue processor, Leica Biosystems, Switzerland), sectioned at 7μm thickness and stained. For morphological analysis, slides were stained with hematoxylin and eosin using Continuous Linear Stainer COT20 (Medite GmbH, Burgdorf, Germany). The outer hair cells and spiral ganglion cell density in Rosenthal's canal of the basal turn were counted. The stria vascularis thickness of the basal turn was measured. To identify goblet cells, the tissue sections were stained with Alcianblue ph 2,5 P.A.S. kit (Bio-Optica Milano, Milano, Italy) according to company's protocol. Tissue sections were imaged using Olympus IX83 widefield microscope with Olympus CellSens (Olympus AG, Volketswil, Switzerland).

## Histopathology of middle and inner ears

A total of 76 out of 88 ears were examined by using a modified pathological scoring system [23]. The score evaluates pathological changes such as middle ear effusion, inner ear effusion, inflammatory cell infiltration, tissue proliferation, tissue debris and goblet cells. This simplified pathological scoring system consist of 4 grades ranging from 0 to 3 indicating pathology absent, very scarce, span part of the middle or inner ear and span the entire middle or inner ear, respectively. The score values are summed up to a final total score for each organ. This score system facilitates the comparison between histopathology changes with ABR data. In this way, mice with highest score value is showing severe pathological changes in the inner ear.

**HC counting.** We performed a relative hair cell count and focused on basal turn of each sample. Cell counting were assessed on three sections obtained from serially cut of a paraffin block (every third 7μm sections). Hair cells were counted and set in relation to the maximum count of hair cells possible, gaining values from 0/9 to 9/9 for OHC and 0/3 to 3/3 for IHC.

SG counting.   We counted spiral ganglion neurons in basal of each sample. The SG cells were counted in an area of 10'000 μm$^2$ as described by Oishi N, et al. [24]. Cells were included, if they met following criteria: cell diameter of 12–15 μm and nucleus diameter of 5–9 μm.

SV thickness.   We measured the thickness of stria vascularis in basal turn of each sample. The thickness of SV was assessed on three sections obtained from serially cut of a paraffin block (every third 7μm sections).

## Statistical methods

To analyze the difference between two groups we used either two-tailed student's t-test (ABR thresholds) or Mann-Whitney test (Score pathology). To analyze the difference between three groups we used either two-way analysis of variance (ANOVA) followed by Tukey's multiple comparison test or Kruskal Wallis test followed by Dunn's multiple comparison test (Graph-Pad PRISM, GraphPad software, La Jolla, CA, USA). P-values <0.05 were considered statistically significant.

## Results

### STAT1 ablation induces uni- and bilateral deafness in mice

We investigated in the absence of STAT1 the hearing ability by measuring the ABR threshold. Mice of different genotypes and ages (3-, 9-, 12–13 and 18-week-old groups) were evaluated (Fig 1A). In the group of the youngest mice (3 weeks old) in our study, the ABR thresholds between wild type (n = 2 ears) and heterozygotes (n = 14 ears) mice were similar. In the group of young mice at 9 weeks of age (wild type n = 4, heterozygotes n = 8, homozygotes n = 8 ears), two from four heterozygous mice showed hearing impairment, one showed a bilateral and the other unilateral phenotype. Significant differences in ABR thresholds were obtained between wild type and heterozygous mice (p = 0.035). In the case of homozygous mice, the average of thresholds was not different between wild type and homozygote, however one mouse (45 dB SPL) out of four mice presented higher hearing thresholds than the wild type (Table 1). The next group of mice aged 12–13 weeks presented differences in ABR thresholds between wild type and knockout mice. Three out of four homozygote mice showed unilateral hearing loss and one bilateral phenotype. The mean of the ABR threshold of wild type mice (n = 8 ears) was significantly lower than those from STAT1-KO (n = 8 ears, p = 0.003). In the group of 18 weeks old mice, the mean value of ABR thresholds was not different between wild type mice and STAT1-KO mice. However, we observed a group of heterozygous and homozygous mice showing high hearing thresholds. Of seven heterozygote mice, three showed bilateral phenotype ranging from profound (> 80 dB SPL) to middle hearing impairment (65 dB SPL) and one unilateral phenotype (50 dB SPL). Of six homozygous mice, one showed bilateral hearing loss (> 80 dB SPL) and two showed unilateral phenotype (65–70 dB SPL). In addition, one out of five wild type mice showed bilateral hearing loss (50–60 dB SPL).

The mean latencies and standard deviations of all ABR waveforms for each genotype group are displayed in Fig 1B. The ABR latencies at 80dB SPL for all waves were similar among the genotypes in the different age groups. However, the oldest group of 18 weeks presented significant differences between genotypes for ABR waves IV and V.

### STAT1-KO mice spontaneously develop chronic inflammation with effusion in the middle ear

To further investigate the cause of hearing impartment in STAT1-KO mice, we examined the anatomy and histology of 76 temporal bones after ABR measurements. All wild type mice

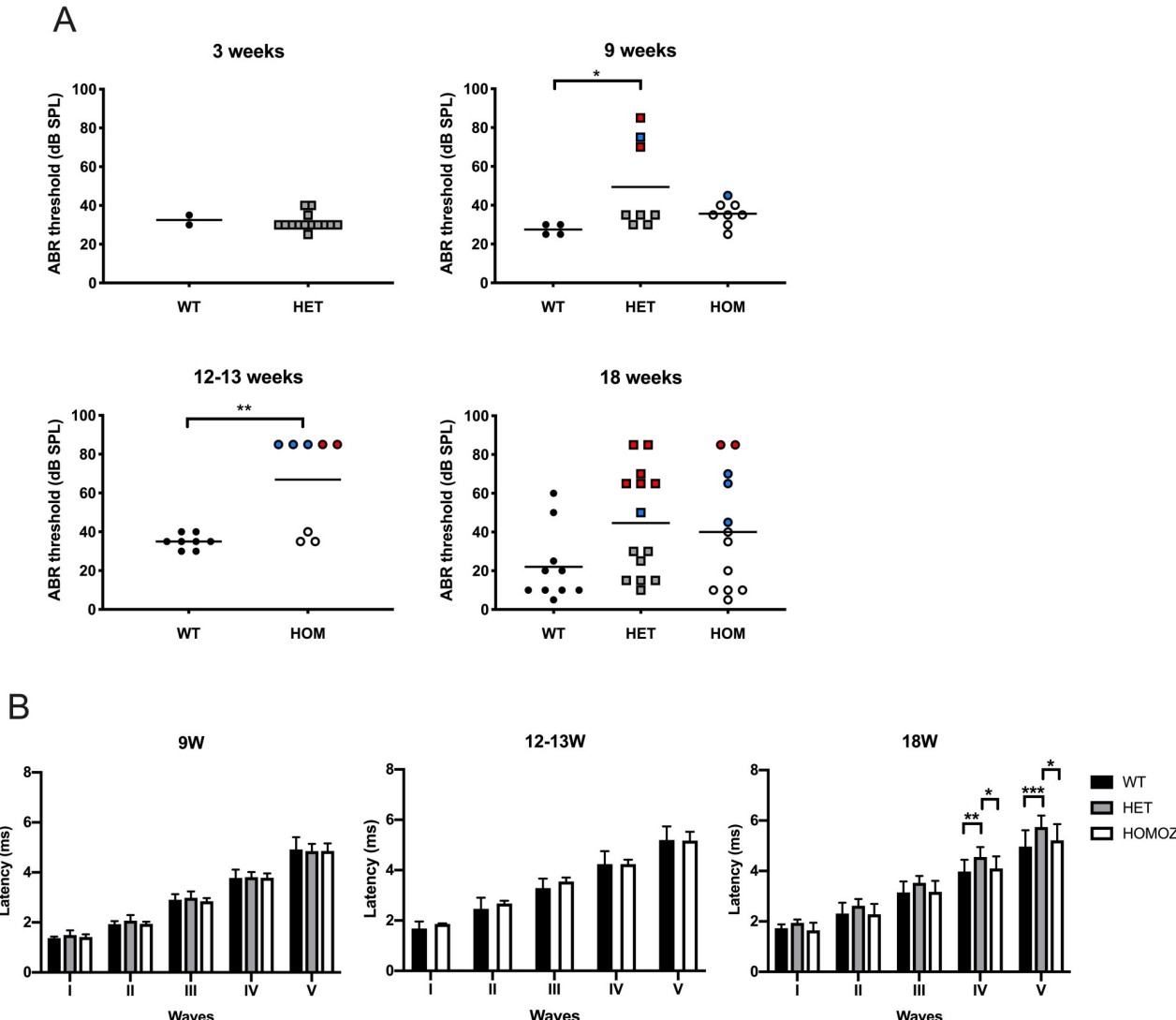

**Fig 1. STAT1-KO mice exhibited unilateral and bilateral hearing deficits.** (A) Analysis of click-induced auditory brainstem response (ABR) of wildtype (WT) and STAT1-KO mice at different age. Each data point represents the click evoked ABR threshold in one ear. Bilateral and unilateral hearing impairment are labeled in red and blue, respectively. Student's *t*-test was used to compare two genotype groups. Kruskal Wallis test followed by Dunn's multiple comparison test was used to compare three groups. (B) The latencies of all five ABR waveforms for 80 dB SPL click stimulus. Two-way ANOVA followed by Tukey's multiple comparison test. Bars represent mean + SD. *$P<0.05$, **$P<0.01$, ***$P<0.01$. HET, heterozygous; HOM or HOMOZ, homozygous.

**Table 1. Number of mice examined.**

| Age (weeks) | Genotypes | | |
|---|---|---|---|
| | Wild type (affected/total) | Heterozygous (affected/total) | Homozygous (affected/total) |
| 3 | 0/1 | 0/7 | - |
| 9 | 0/2 | 2/4 | 1/4 |
| 12–13 | 0/4 | - | 4/4 |
| 18 | 1/5 | 4/7 | 3/6 |

The affected mice included those mice showing unilateral and bilateral hearing impairment.

presented normal tympanic membrane, normal round window membrane and clear middle ear cavity (MEC) with thin epithelium, except from one mouse of 18 weeks old with normal hearing. This mouse presented in one ear a middle inflammation with light middle and inner ear effusion as well as tissue proliferation. STAT1-KO mice of 9 and 13 weeks with unilateral or bilateral hearing impairment presented from moderate to high levels of inflammation in the affected ear. While those mice with normal hearing did not show signs of inflammation. STAT1-KO mice at 18 weeks presented different situations. All heterozygous mice with high ABR thresholds presented sign of inflammation, except one mouse that presented bilateral hearing loss with one ear affected and the other ear with almost no inflammation. Two homozygous mice with unilateral hearing impairment did not present signs of inflammation in the affected ears. All other STAT1-KO mice with high ABR thresholds presented signs of inflammation, while those mice with normal ABR threshold showed no inflammation signs. Examples of middle ear inflammation in STAT1 mice of all groups are shown in Fig 2. STAT1-KO mice exhibited thickened and retracted tympanic membrane, thickened round window membrane accompanied with exudate as well as inflammatory cell infiltration in MEC with exudate (Figs 2 and 3).

Typical inflammatory changes in the MEC of affected STAT1-KO mice included suppurative exudates, inflammatory cells, mucosal thickening, fibrous polyps and variable severity of epithelial hyperplasia (Fig 3). In addition, those deaf STAT1-KO mice presented chronic inflammation with proliferation of connective tissue and blood vessels. Most observed inflammatory cells in our samples included macrophages, lymphocytes, eosinophils and few neutrophils (Fig 3L). Moreover, mucins produced by goblet cells were found in all mice genotype of all age groups. While individual goblet cells were present in the middle ear epithelium of wild type, goblet cell clusters were present in the epithelium of affected STAT1-KO mice (Fig 4).

## ABR thresholds increase with score pathology

We investigated the correlations between the ABR thresholds and the score pathology independent of the mice genotype. The degree of inflammation on histopathology was graded using a scoring system ranging from 0 (no inflammation) to 3 (severe inflammation) as described in material and methods section. The scoring system evaluated the extent of pathological changes in the middle and inner ear such as middle ear effusion, inner ear effusion, inflammatory cell infiltration, tissue proliferation, tissue debris and clusters of goblet cells. The score values were summed up to a final total score for each organ. Individual samples with high ABR thresholds are located further to the right and those with high pathology scores are further up on the graph (Fig 5A). The ABR thresholds and the final total pathology scores for all examined groups have shown strong positive linear relationships achieving correlation coefficients of 0.9, 0.94 and 0.74 for 9, 12–13 and 18 weeks old mice, respectively. This indicated that the ABR thresholds increased as did the score pathologies.

We next investigated, if the score pathology changed between the genotypes. There were no differences in pathology scores between the genotypes except the significant difference that has been found in the group of 12–13 weeks old mice (Fig 5B). However, high pathology scores of more than 5 have been observed in individual heterozygous and homozygous mice in all other age groups.

## STAT1-KO mice show thicker SV than those of the wild type mice, but show similar numbers of HC and SG cells

We further assessed changes in inner ear structures such as number of hair cells, spiral ganglion neurons and thickness of stria vascularis. We did not observe any structural changes

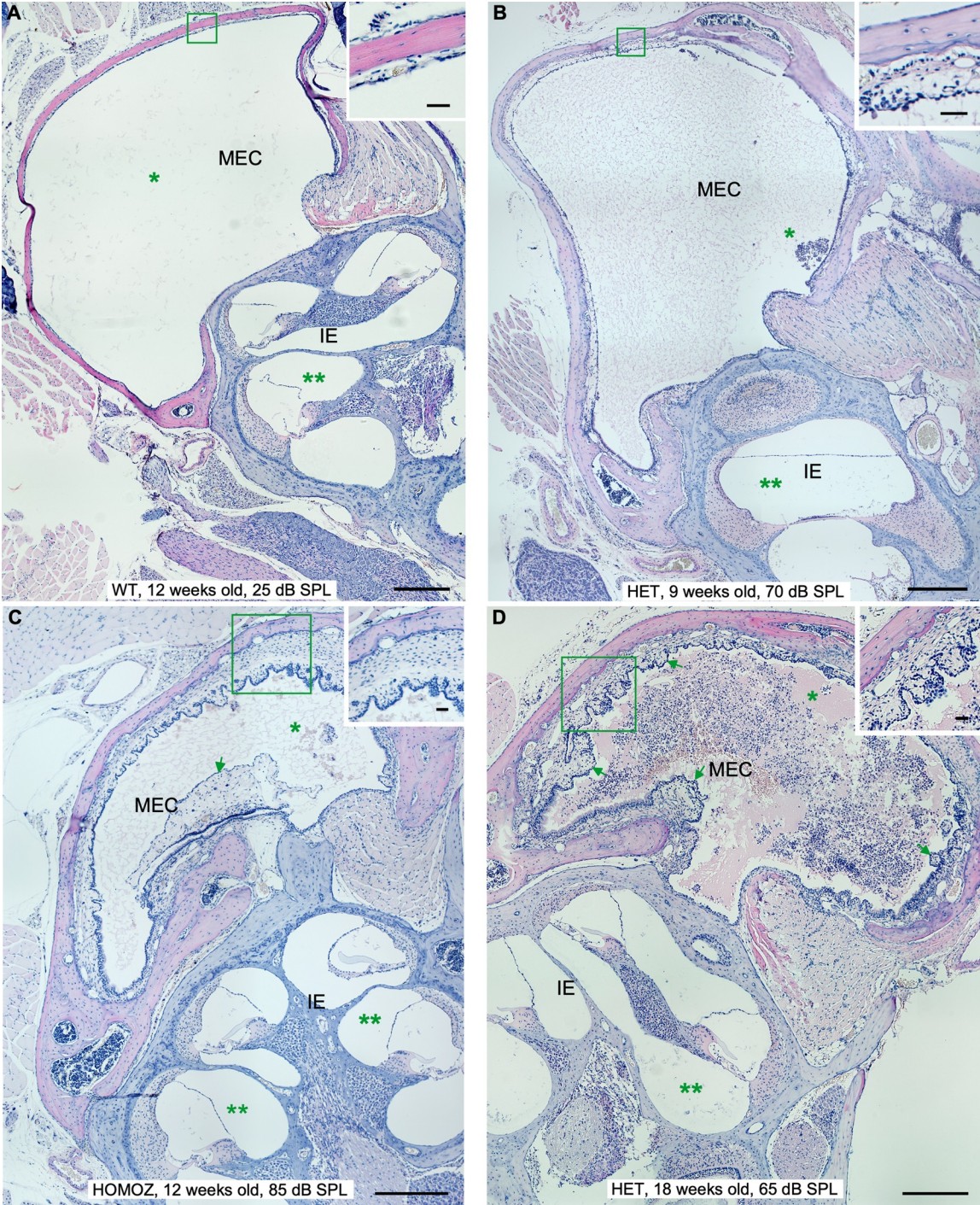

**Fig 2. Manifestations of middle ear inflammation in STAT1-KO mice at different ages.** From mild to chronic inflammation degrees in middle ear and cochlea. (**A**) Wild type (WT) mice at 18 weeks of age, heterozygous (HET) STAT1-KO mice at 9 (B), homozygous (HOMOZ) mice at 12 (C) and heterozygous (HET) mice at 18 (D) weeks of age. Clear appearance of middle ear cavity (MEC) of wild type in comparison to MEC of knockout mice filled with watery effusion with few inflammatory cells (asterisk in B, C) and large number of inflammatory cells (asterisk in D). The cochlea showed no effusion in wild type (two asterisks in A) or effusion with few inflammatory cells in knockout mice (two asterisks in D). The thickness of the middle ear epithelium is greater in knockout mice (insets). Additional signs of inflammation in knockout mice included fibroblastic tissue (arrow in C) and fibrous polyps (arrows in D). Scale bars 300 μm. Inset scale bars 30μm. CO, cochlea; EAC, external auditory canal.

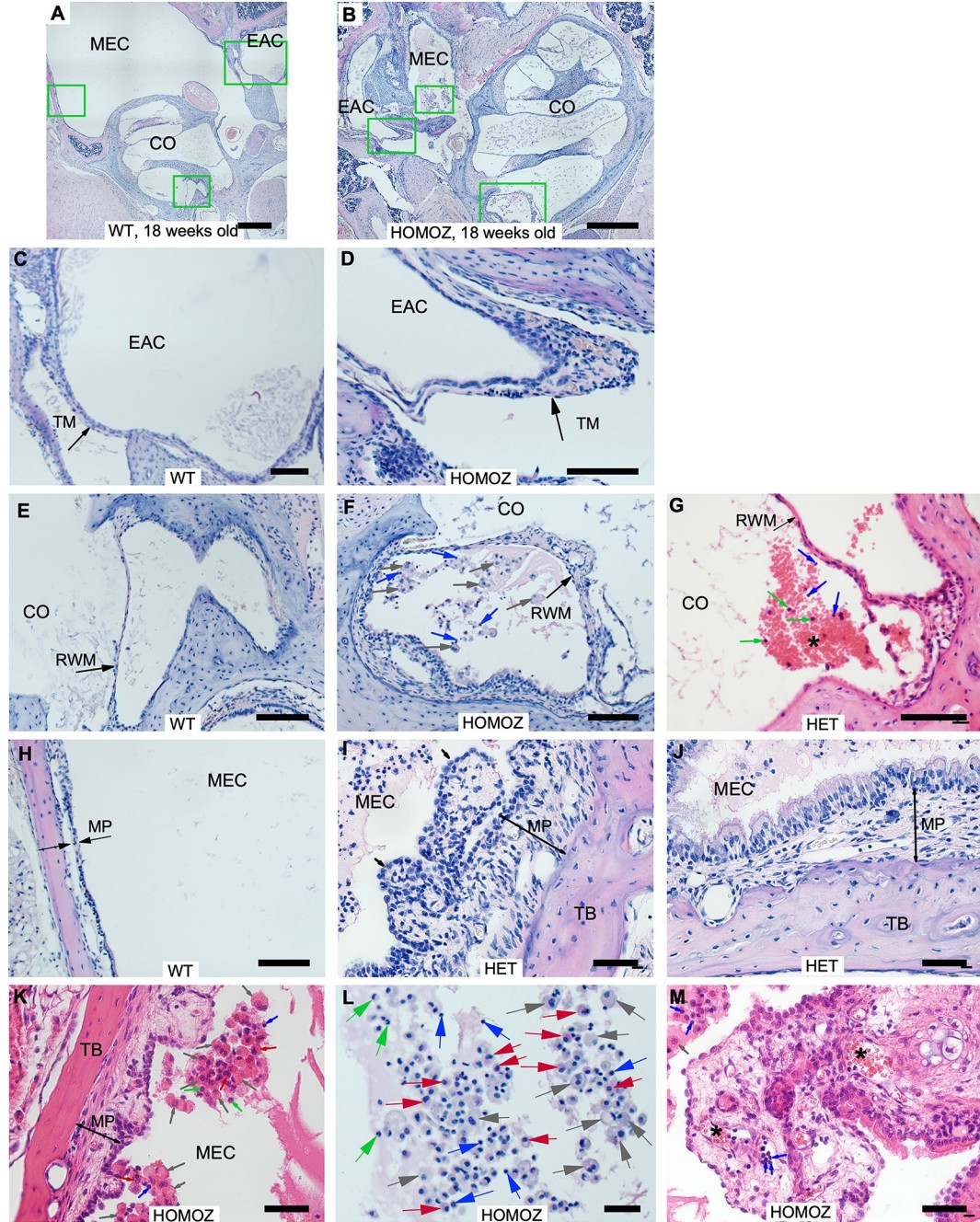

**Fig 3. Morphological changes of STAT1-KO mice.** (A-B) Representative images from middle ears of 18 weeks old wild type and STAT1-KO mice. Scale bars 500µm. (C, E, H) High magnification images of wild type mice (A). (D, F, L) High magnification images of STAT1-KO mice (B). Scale bars in D and F 100µm, in L 10µm. (C, D) The thickness of the tympanic membrane (TM) was greater in STAT1-KO mouse compared to wild type mouse. (E-G) The round window membrane (RWM) of STAT1-KO was thicker than the membrane of wild type mice. (H-K) The thickness of mucoperiosteum (MP) of knockout mice containing epithelial hyperplasia and fibrous polyps (black arrows) was much greater compared to MP of wild type mice. Scale bars in I, J, K 50µm. Different manifestations of inflammatory cell infiltration were observed in STAT1-KO mice (F, G, I, J, K, M). (G) In addition, cluster of erythrocytes (*) was also detected. Scale bar 50µm. (L) Representative image of a middle ear cavity (MEC) of knockout mouse that was partially filled with exudate comprising inflammatory cells such as macrophage as single or multinucleated (grey arrow), lymphocytes (blue arrow), neutrophil (green arrow), eosinophil (red arrow). (M) MEC from a mouse showing disruption of ciliated epithelium with fibrous proliferation and clusters of lymphoid infiltrates (arrows) and cluster of erythrocytes (*). Scale bar 50µm. CO, cochlea; HET, heterozygous; HOMOZ, homozygous; TB, temporal bone; WT, wild type.

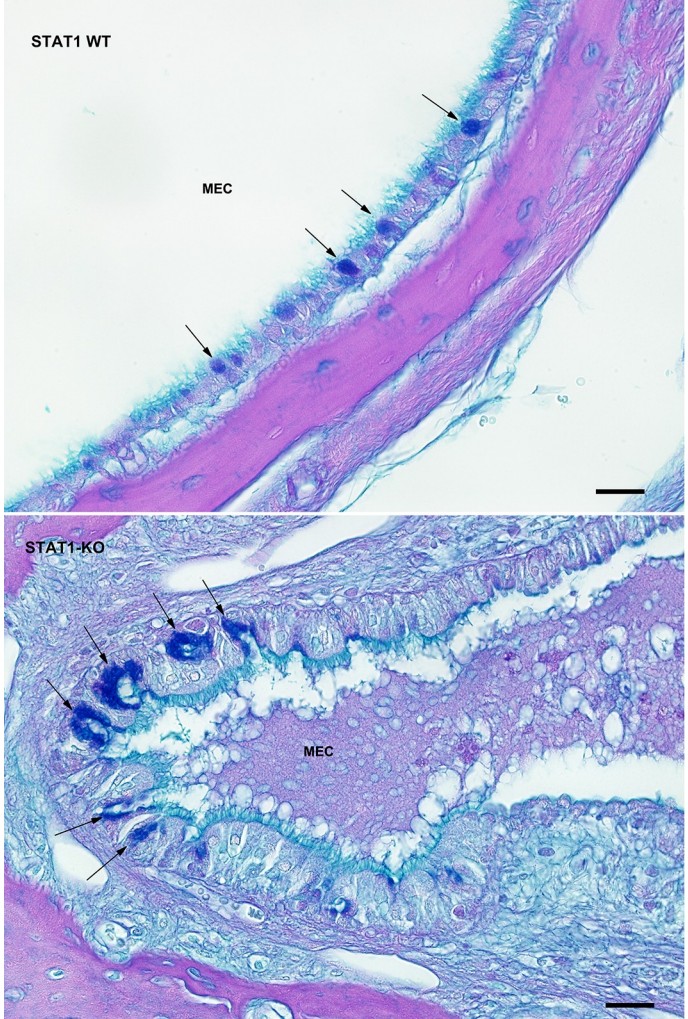

**Fig 4. Cluster of goblet cells were found in STAT1-KO mice with otitis media.** Middle ear sections from wild type and STAT1-KO mice were stained for goblet cells by mucin staining using AB-PAS method. More AB-PAS-positive cells were found in the mucosal layer of the MEC of STAT1-KO mice than in wild type (WT), as indicated by arrows. Scale bars 30 μm.

between the genotypes of STAT1, except for stria vascularis thickness. The analysis of the thickness of the stria vascularis revealed a significant difference between the STAT1-KO (22.5 ±2.1) and wild type (20.6±1.4) mice in the basal turn of the cochlea (p<0.05; Fig 6).

## Discussion

We report that mice with ablation of the STAT1 gene regularly develop otitis media without any experimental interventions and even though those mice were housed in pathogen-free conditions. No previous studies have reported spontaneous inflammation in STAT1-deficient mice, instead several studies challenged these knockout mice with different pathogens to investigate the role of STAT1 as intermediate player in the immune system [17, 18]. Patients with partial or complete loss of STAT1 functions are susceptible to infection suggesting that the actions of STAT1 are indeed essential in the inflammatory responses of IFN to pathogen infection [17, 19–21].

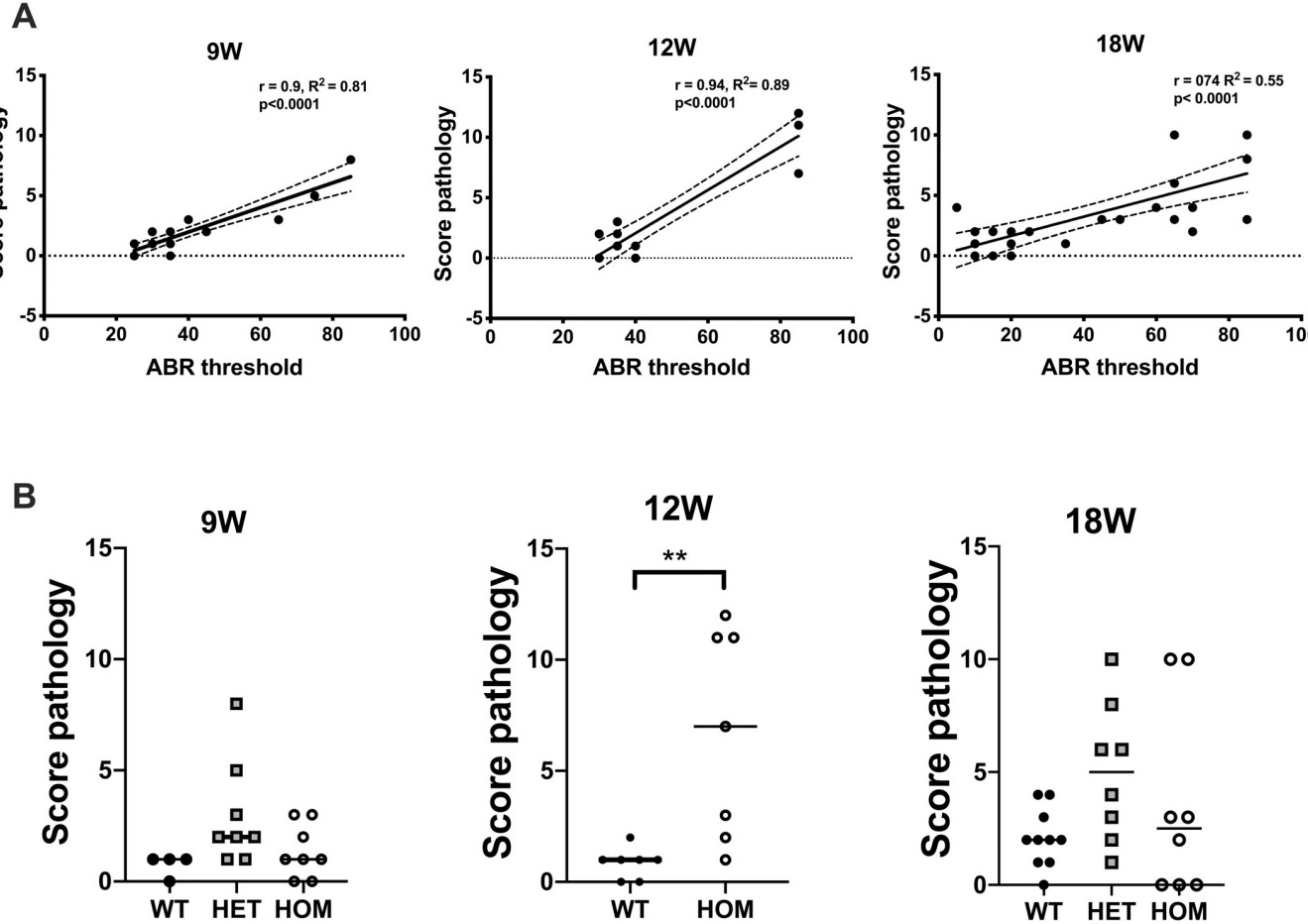

**Fig 5. ABR thresholds increase with the score pathology.** (A) Correlations between ABR thresholds and score pathologies. Mice ears with a higher ABR thresholds are associated with higher score pathology. Growth slope was calculated using a linear regression. Dashed lines depict the 95% confidence interval. (B) STAT1-KO mice at 12 weeks of age exhibit higher score pathology than wild type (WT) mice. Each data point represents the sum of the pathology score in one ear. Mann-Whitney test was used to compare two genotype groups. Kruskal Wallis test followed by Dunn's multiple comparison test was used to compare three groups. **$P<0.01$. HET, heterozygous; HOM homozygous.

We observed hearing impairment in our mice at the age of 9 weeks and not at the age of 3 weeks. Although the ABR thresholds were low and similar in wild type and heterozygous mice at 3 weeks of age; we cannot rule out that homozygous mice might have showed hearing impairments. The timing of the elevation of ABR thresholds seems to differ between knockout mice. Tian *et al*. found that *Enpp1asj/asj* mice exhibited normal ABR thresholds at 3 weeks and higher threshold by 6 weeks of age [3]. While C3H/HeJ mice presented normal thresholds at 3 months and started to show elevated thresholds at 5 months [9]. The *Tgif* mutant mice at 2 months of age showed higher average ABR thresholds as compared to wild type. Already at the age of 1 month, the homozygous *Tgif* mice presented reduced Preyer's reflex [8].

The degrees of hearing impairment in our knockout mice were very variable between the age groups and genotypes. Mice of 12–13 weeks and heterozygous mice of 9 weeks displayed significantly higher hearing thresholds compared to wild type. However, the average of ABR thresholds between the genotypes of other mice groups did not always reach significance. When we further analyze individual mouse in each group, we observed that there was an increased incidence of hearing impairment in knockout mice than in the wild type mice. All STAT1-KO animals of 12–13 weeks presented hearing impairment suggesting a 100%

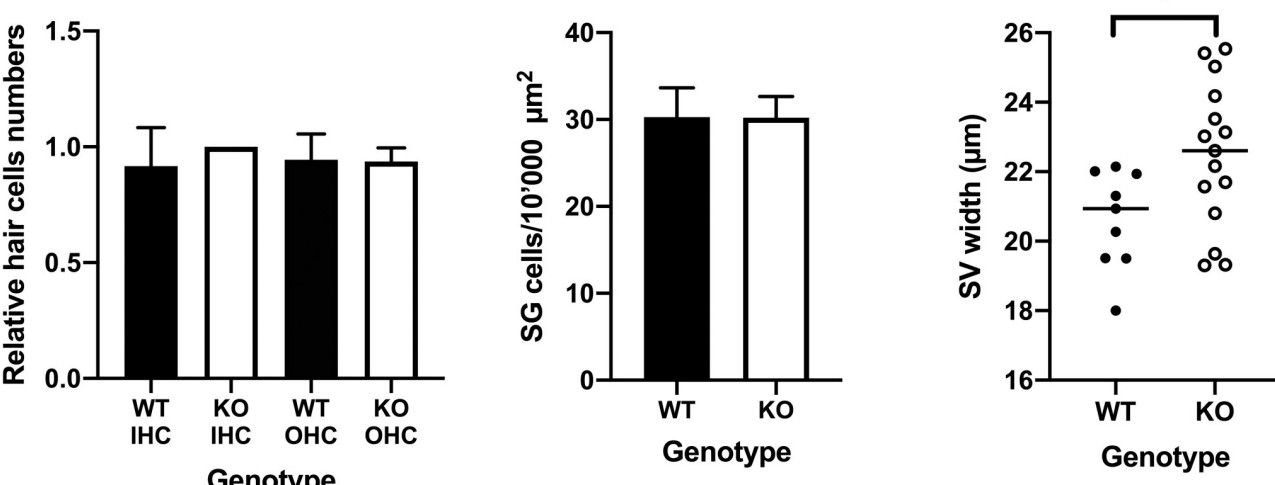

**Fig 6. No loss of HC and SG cells in 12–13 weeks old mice.** (A) Comparison of the relative cell numbers of inner hair cells (IHC) and outer hair cells (OHC) between wild type (WT) and STAT1-KO mice (KO). (B) Comparison of spiral ganglion (SG) cells between WT and STAT1-KO mice. (C) Comparison of the stria vascularis (SV) thickness between WT and STAT1-KO mice. STAT1-KO mice show slight thickener membrane as those from the WT mice. Two tailed student's *t*-test was used to compare two genotype groups. *P<0.05

incidence, while STAT1-KO mice of 9 and 18 weeks showed hearing impairment in 3 out of 8 and 7 out 13, respectively, suggesting a moderate incidence ranging from 37.5–54%. Similar high incidence of hearing loss of 90% has been also found in 12 weeks old *Enpp1^{asj/asj}* mice [3]. We might expect that the incidence of hearing impairment increases with age as previously shown [25]. The ABR thresholds in C3H/HeJ and *Enpp1^{asj/asj}* mice also increased with age [3, 9]. However, it was not always the case with the STAT1-KO mice. From our histopathology data, the distribution of cases with spontaneous inflammation in STAT1-KO mice between the age groups was variable. Moreover, it seems that spontaneously resolution of inflammation occurred in some STAT1 mice. These findings suggest that at the time point of ABR measurements, the knockout mice of 12–13 weeks were in a stage of chronic inflammation, while the other mice groups were at different stages of inflammatory process or resolved inflammation. Further studies are needed to clarify the role of STAT1 on hearing using the same experimental group of mice over a long period of time. The case of the single wild type mouse of 18 weeks old with hearing impairment observed in our study might be a case of inborn hearing impairment or acquired later in life.

The strong positive correlation between ABR thresholds and pathology scores suggested that these variables are associated and increased together. Similar correlations between ABR elevation and presence of middle ear inflammation had been reported [3, 4, 8, 9]. The correlation coefficient of the group of 18 weeks was still strong value, but it was weaker than those obtained from the other groups of 9 and 12–13 weeks. Animals with low levels of ABR thresholds presented mostly low score pathologies as we might expect. While animals with high hearing thresholds presented mostly high score pathology. However, three mice presented either moderate or high ABR thresholds and low score pathology. In the presence of these outliners the strength of the correlation in the group of 18 weeks was weaker in comparison to the two other groups. These three outliners in the region of high ABR thresholds belong to heterozygous and homozygous mice. Although we did not detect signs of inflammation, the resulting hearing impairment may indeed be the results of interaction of multiple factors, such as genetic background and aging processes. It has been reported that the C57BL/6J mice at the

age of 3–6 months start to show elevated hearing thresholds at high frequencies [25]. During the aging processes several changes continuously happen affecting the normal functions in the body in much as in the hearing. This could explain the high ABR thresholds observed in those animals without signs of inflammation. We may therefore expect that the observed correlation will get weaker with the age. Another interesting observation was the presence of one wild type mouse with normal hearing and middle inflammation in one ear. The inflammation in this mouse might have been of short duration or spontaneously resolved and its magnitude was not sufficient to destroy the structure of the auditory organ at the time of ABR measurements.

Genotype differences had been observed in *Tgif* mutant mice, while homozygous mice demonstrated high incidence of otitis media, heterozygous revealed chronic otitis media with low penetrance [8]. In our study, difference in average pathology scores between genotypes was visible in the group of 12–13 weeks. While the average between the genotypes in the other age groups did not reach significance. The pathology scores were scattered and shown more variability then those of the wild type mice. The scattering values observed in knockout mice might reflect different stages and magnitudes of inflammation. Otitis media often resolves spontaneously, but recurrent and otitis with effusion are longer lasting; and thereby can causes more damage in the ear. Barlett *et al.* suggested that *Splunc1⁻/⁻* mice experienced recurrent episodes of otitis as they observed that the MEC underwent frequent remodeling [6]. The mice groups in this study may indeed contain individuals with spontaneous resolved inflammation that improved histopathology and allowed partial recovering of hearing. While those mice with long period of inflammation might progressively be affected leading to irreversible damage and permanent hearing loss. The STAT1-KO mice seem to be susceptible to otitis media, but we cannot exclude that other factors contribute to inflammation in mice, as the otitis media is a multifactorial disease [26].

Early study reported that STAT1-KO mice presented normal production of B-lymphocytes and monocytes in fetal liver as well as the production of T-lymphocytes in neonatal thymus [16]. Moreover, Lee *et al.* reported that lymphocytes of STAT1-KO mice showed enhanced survival and proliferation *in-vitro*, while normal lymphoid homeostasis in STAT1-KO mice was observed [27]. In our study, STAT1-KO mice showed variable degree of inflammatory cell infiltration. Our findings may reflect also an enhanced survival of lymphocytes in those middle ears of STAT1-KO mice with high degree of infiltration. This could be the result of missing immune modulatory activities of STAT1. Similar findings of variation of inflammatory cell content in *Enpp1^{asj/asj}* mice had been reported [3]. Interestingly, patients with mutations in STAT1 gene are susceptible to infections including ear infections [17, 19, 21]. Toubiana *et al.* reported patients suffering chronic mucocutaneous candidiasis (CMC) with gain of function (GOF) mutations in STAT1, those patients did not display relevant defects in immune parameters except but low memory B cells and low production of IL-17A⁺ T cells after pathogen stimulation [21]. In addition, Sharfe *et al.* have reported that patients with heterozygous mutations presented progressive loss of T-lymphocytes and reduced STAT1 expression [20]. In contrast, Mizoguchi *et al.* investigated a group of CMC patients and found that 50% patients carried STAT1 GOF mutations. Those patients did not present progressive lymphopenia, but a broad infectious phenotype [19]. It seems that the regulation of an optimal STAT1 activity is important and dysregulation of its activity leads to immune system imbalance.

STAT1 is involved in the regulation of immune system and in the process of tumor formation. Persistent activation of STAT3/STAT5 often promotes chronic inflammation leading to transformation of healthy cells into malignant cells, while STAT1 suppress tumor growth. Inflammation and cancer are associated processes; inflammatory cells regulate cancer progression and form tumor environment besides enhanced cell proliferation and survival [28]. In our knockout model, inflammation was manifested as hyperplasia of epithelial cell, goblet cells

and lymphocytes infiltration. Lee *et al.* reported that STAT1-KO mice showed increased susceptibility to thymic tumor induction [27]. The lack of STAT1 predisposed the mice to increased incidence of colorectal cancer showing epithelial cell proliferation and decreased apoptosis during the cancer development [29]. We could hypothesize that inflammatory responses observed in the middle and inner ear of our knockout mice could be due to enhanced expression of STAT3/STAT5 in absence of STAT1 as observed in cancer cells [28]. The immune modulatory activities of STAT1 were missing and this may cause an uncontrolled infiltration of inflammatory cells in the middle ear. Moreover, the disturbance in the critical balance between pro-survival and pro-apoptotic proteins leading to reduced apoptosis could be due to the lack of apoptotic function of STAT1 as reported for other genes involved in inflammation such as *TNF-α*, *Tgif1* and *Tlr4* [2, 8, 9]. Further research will be necessary to clarify this point.

One of the weakness of our study is that the progress of hearing impairment with age and the recurrence of the otitis were not examined. Each group of mice examined was immediately euthanized after completion of ABR measurements. Future studies of STAT1-KO mice at various inflammation stages would clarify the development of inflammation process and the recruitment time of the inflammatory cells.

## Conclusions

STAT1-KO mice do not show any obvious structural deformations, but presented increased ABR thresholds suggesting that otitis media could be the major cause of hearing impairment. Our findings may also suggest that the absence of STAT1 is associated to uncontrolled proliferation of epithelial cell of the middle ear cavity and disturbance in infiltration of inflammatory cells. This suggests that loss of STAT1 predispose the mice to spontaneous otitis media.

## Acknowledgments

We thank the Microscope core Facility of the Department of Biomedicine for training, advice, and use of the microscope. We also thank Dominik Viscardi and members of the mouse facility for assistance in animal care.

## Author Contributions

**Conceptualization:** Daniel Bodmer, Soledad Levano Huaman.

**Data curation:** Peter Kern, David Bächinger, Arianne Monge Naldi, Soledad Levano Huaman.

**Formal analysis:** Peter Kern, David Bächinger, Arianne Monge Naldi, Soledad Levano Huaman.

**Funding acquisition:** Daniel Bodmer.

**Investigation:** Peter Kern, Soledad Levano Huaman.

**Supervision:** Daniel Bodmer, Soledad Levano Huaman.

**Writing – original draft:** Soledad Levano Huaman.

**Writing – review & editing:** Daniel Bodmer, Arianne Monge Naldi, Soledad Levano Huaman.

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
