## [Decision Letter · Decision Letter 0]

10 Jul 2020

PONE-D-20-17146

STAT1 deficiency predispose to spontaneous otitis media

PLOS ONE

Dear Dr. Levano Huaman,

Thank you for submitting your manuscript to PLOS ONE. After careful consideration, we feel that it has merit but does not fully meet PLOS ONE’s publication criteria as it currently stands. Therefore, we invite you to submit a revised version of the manuscript that addresses the points raised during the review process.

The reviewers have made several comments regarding the results and discussion of the manuscript that should be further addressed. More importantly, the "discussion" should focus on the most prominent findings of the manuscript rather than discussing the already existing literature in this regard.

We look forward to receiving your revised manuscript.

Kind regards,

Rafael da Costa Monsanto, M.D.

Academic Editor

PLOS ONE

Journal Requirements:

2. Please include the source of the animals in the Methods section, in accordance with PLOS ONE requirements for reporting and reproducibilty.

Reviewers' comments:

Reviewer's Responses to Questions

**Comments to the Author**

1. Is the manuscript technically sound, and do the data support the conclusions?

Reviewer #1: Yes

Reviewer #2: Partly

2. Has the statistical analysis been performed appropriately and rigorously? 

Reviewer #1: Yes

Reviewer #2: No

3. Have the authors made all data underlying the findings in their manuscript fully available?

Reviewer #1: Yes

Reviewer #2: Yes

4. Is the manuscript presented in an intelligible fashion and written in standard English?

Reviewer #1: Yes

Reviewer #2: No

5. Review Comments to the Author

Reviewer #1: This is a very nicely presented manuscript with excellent basic science and clinical significance. The discovery that STAT1 deficiency could be a factor in otitis media is important for clinicians seeking to determine factors associated with otitis media in patients.

ABR data are nicely displayed and discussed. Histopathology samples are of high quality and support the conclusions of the authors.

Some minor questions should be addressed by the authors:

Why were the ABR thresholds better in homozygous mice at 18 weeks compared to those obtained at 12-13 weeks? Do the authors think that the inflammation had spontaneously resolved, allowing partial recovery of hearing?

Why was the score pathology lower in homozygous STAT1 deficient mice at 18 weeks compared with 9 and 12 weeks? Do the authors think that the inflammation had spontaneously resolved resulting in improvement in histopathology?

Also, English grammar needs some minor corrections.

Reviewer #2: STAT1 deficiency predispose to spontaneous otitis media.

In this manuscript, Bomer et al, investigate the consequences of STAT1 deficiency on hearing and inflammation in the inner and middle ear. Using STAT1+/- and STAT1 knockout mice previously characterized, they examined hearing abilities by measuring auditory brainstem response (ABR) at different ages. They also investigate the ear histology, histpathology, as well as structural abnormalities in STAT1 KO mice. They conclude that STAT1 deficiency increases susceptibility to otitis media through spontaneous inflammation.

Overall, the manuscript will benefit from a careful editing of introduction and results sections. Discussion needs to be rewritten to highlight the findings from the current study. Comments are detailed below:

Major points:

- In line 86 of Material and Methods section, the authors wrote: “ABR thresholds were measured in STAT1 mice aged 3 weeks (WT, n=1; heterozygote, n=7)”. However, in line 160 of the Results section, they mention: “ABR thresholds between WT (n=4 ears) and heterozygotes (n=14 ears) mice were similar.” Can authors clarify the number of WT mice used at 3w?

- Line 241: “The onset of otitis media seems to start later than 3 weeks as the hearing impairment has not been observed at this young group of mice.” Comparing the ABR measurements in only one WT mouse to 7 heterozygotes is not sufficient to make this conclusion. Can authors provide ABR measurements for more WT mice at 3 weeks?

- Figure 1: At 9 weeks, homozygotes only have a moderate hearing loss and significant hearing impairment is only observed in heterozygotes. Whereas at 12-13 weeks, homozygotes have a significant increase in ABR thresholds. At 18 weeks, there is no significant difference in ABR thresholds between WT, heterozygotes and homozygotes. Can authors discuss this observation and its significance? Do they expect similar ABR thresholds at 18 weeks?

- Can authors comment on the absence of homozygotes in 3 weeks, as well as heterozygotes in 12-13 weeks?

- Line 185: “STAT1-KO mice presented from moderate to very significant degree of inflammation in the middle and inner ear at 9, 12-13 and 18 weeks mice.” In figure 1A, at 9 weeks there is no difference in ABR measurements between homozygotes and WT. Can authors discuss why they observe an inflammation without an impairment in hearing?

- Can the authors mark the area with differences between KO and WT on Figure 2. Also, it will make it easier for the readers to have the panels labeled with genotypes, age, as well as high or low ABR thresholds.

- Line 210: “The final total pathology scores for all examined groups (9, 12-13 and 18 week old mice) have shown good correlation with ABR thresholds”. Please refrain from using terms such as “good

correlation”, and report the correlation coefficient number instead. The correlation and whether it is meaningful can be discussed in the discussion section.

- Line 221: “We didn’t observe any structural changes between the genotypes of STAT, except for the stria vascularis. The analysis of the thickness of the stria vascularis revealed a significant difference between the STAT1-KO and WT mice in the basal turn of the cochlea (Fig 6).” Please report the SV width number and p value in the text as well as in figure 6.

- Discussion needs to be reorganized and focus on what the authors have found, rather than going through what has been previously published. For example, lines 231 to 241, 242 to 247, 257 to 281, and 284 to 289, are summarizing findings from previous studies. This information would be better placed in the introduction section to give readers context as to the rationale of the manuscript. Authors should discuss their own results in the Discussion section and argue the significance of their findings.

- Results from this manuscript and previous studies suggest that hearing impairment can be detected from 9 weeks onwards. Can authors discuss why the lowest R2 is observed for STAT1 KO mice at 18 weeks, rather than earlier ages (Figure 5A)? Did they expect the lowest correlation at this age?

- Line 214: “There were 215 no differences in score pathology between the genotypes except from the group of 12-13 weeks old mice.” The authors need to discuss the significance of this observation in the Discussion section.

- The article by Mizoguchi et al, 2014 (PMID: 24343863) needs to be cited and authors need to discuss the results from this article in the context of their findings in the discussion.

- Examining inflammatory cells such as lymphocytes and macrophages in only conducted for STAT1 KO mice (Figure 3). These experiments need to be performed including WT mice side by side for comparison.

- Both gain and loss of function of STAT1 are linked to otitis media (Toubiana et al, 2016 and Sharfe et al 2014). Patients with heterozygous variants in STAT1 present immunodeficiency with progressive loss of lymphocytes, in contrast to what authors have described here. Also, it is reported that siRNA mediated knockdown of STAT1 reduces inflammatory mediators and can be used to treat ototoxicity (Kaur et al, 2011). Can the authors comment on whether this could be a dosage dependent effect? And how their findings relate to previous reports?

- Line 281: “We did not find any tumor in the middle ear of our knockout mice, however STAT1-KO mice presented signs of inflammation.” Where are the experiments supporting their argument on tumors? Authors need to provide these data. These statements need to be included in Results section, and not to be introduced for the first time in Discussion.

- Line 291: “We could suggest that the roles of STAT1 as tumor suppressor and inflammation regulator were evident in these mice.” Authors have not performed any experiments specifically examining

the role of STAT1 in tumorigenesis, and not detecting tumors is not sufficient to make this argument. Authors need to be careful not to over interpret their results.

Minor points:

- Line 294 should read “STAT3/STAT5 in absence of STAT1 as observed in cancer cells.”

- Authors need to maintain a formal tone throughout the manuscript, e.g., line 221, “didn’t” should be replaced by did not.

- Figure 4 caption: Rather than repeating the same sentence as included in Results: “While few goblet cells were detected in WT mice, cluster of goblet cells were observed in STAT1-KO mice.”, authors should more specifically describe what the figure is showing and what arrows are indicating.

6. PLOS authors have the option to publish the peer review history of their article (what does this mean?). If published, this will include your full peer review and any attached files.

Reviewer #1: No

Reviewer #2: No

---

## [Author Response · Author response to Decision Letter 0]

21 Aug 2020

Dear Dr. da Costa Monsanto,

We greatly appreciate your consideration to revise our manuscript. We also thank the Reviewers for their useful comments and valuable suggestions that have greatly improved the quality of our manuscript. We have carefully considered your comments and addressed the Reviewer’s questions and comments in a point by point reply. Please refer to the document "Response to Reviewers".

Based on reviewer’s comments and recommendations, the main changes to the revised manuscript are:

• The introduction section has been edited

• The result section has been edited and adjusted to improve the presentation of our findings

• The discussion section has been completely rewritten. It is now focused on our most prominent findings and provided now more arguments about their significance

• Figures 1, 2 and 3 have been improved

We hope that the revised manuscript will now meet the editorial criteria and will receive your kind consideration.

Sincerely,

Soledad Levano Huaman

---

## [Decision Letter · Decision Letter 1]

16 Sep 2020

STAT1 deficiency predispose to spontaneous otitis media

PONE-D-20-17146R1

Dear Dr. Levano Huaman,

We’re pleased to inform you that your manuscript has been judged scientifically suitable for publication and will be formally accepted for publication once it meets all outstanding technical requirements.

Kind regards,

Rafael da Costa Monsanto, M.D.

Academic Editor

PLOS ONE

Additional Editor Comments (optional):

Congratulations on the excellent piece of work.

Reviewers' comments:

Reviewer's Responses to Questions

**Comments to the Author**

1. If the authors have adequately addressed your comments raised in a previous round of review and you feel that this manuscript is now acceptable for publication, you may indicate that here to bypass the “Comments to the Author” section, enter your conflict of interest statement in the “Confidential to Editor” section, and submit your "Accept" recommendation.

Reviewer #1: All comments have been addressed

Reviewer #2: All comments have been addressed

2. Is the manuscript technically sound, and do the data support the conclusions?

Reviewer #1: Yes

Reviewer #2: Yes

3. Has the statistical analysis been performed appropriately and rigorously? 

Reviewer #1: Yes

Reviewer #2: Yes

4. Have the authors made all data underlying the findings in their manuscript fully available?

Reviewer #1: Yes

Reviewer #2: Yes

5. Is the manuscript presented in an intelligible fashion and written in standard English?

Reviewer #1: Yes

Reviewer #2: Yes

6. Review Comments to the Author

Reviewer #1: (No Response)

Reviewer #2: (No Response)

7. PLOS authors have the option to publish the peer review history of their article (what does this mean?). If published, this will include your full peer review and any attached files.

Reviewer #1: No

Reviewer #2: **Yes: **Bahareh A. Mojarad

---

## [Editor Report · Acceptance letter]

18 Sep 2020

PONE-D-20-17146R1 

STAT1 deficiency predisposes to spontaneous otitis media 

Dear Dr. Levano Huaman:

I'm pleased to inform you that your manuscript has been deemed suitable for publication in PLOS ONE. Congratulations! Your manuscript is now with our production department. 

Kind regards, 

on behalf of

Dr. Rafael da Costa Monsanto 

Academic Editor

PLOS ONE